# Biocompatible Composite Protective Thin Layer Containing Cellulose Fibers and Silica Cryogel

**DOI:** 10.3390/gels11070522

**Published:** 2025-07-05

**Authors:** Marius Horvath, Katalin Sinkó

**Affiliations:** Institute of Chemistry, Eötvös Loránd University, 1053 Budapest, Hungary; marius@student.elte.hu

**Keywords:** biocompatible composite, silica cryogels, sol–gel, cellulose fibers, thermal and electrical protection

## Abstract

The aim of the present research was to synthesize protective composite layers from biodegradable cellulose and biocompatible, sol–gel-derived silica cryogel. An important task in the present work was to achieve good applicability on distinct (smooth and rough) surfaces of various materials (from metallic to ceramic). The aim was to utilize the composite layers as thermal and electric insulation coating. The investigation put some effort into the enhancement of mechanical strength and the elasticity of the thin layer as well as a reduction in its water solubility. The removal of the alkali content leads successfully to a significant reduction in water solubility (97 wt% → 1–3 wt%). Adhesion properties were measured using a specialized measurement technique developed in our laboratory. Treatments of the substrate surface, such as alkaline or acidic etching (i.e., Na_2_CO_3_, HF, water glass), mechanical roughening, or the application of a thin alkali-containing primer layer, strongly increase adhesion. SEM analyses revealed the interactions between the matrix and the reinforcement phase and their morphology.

## 1. Introduction

One of the important challenges of the recent investigation is to develop a thin, environmentally friendly insulation layer for distinct (smooth and rough) surfaces of various materials (from metallic to ceramic). Extensive research has focused on developing advanced insulating materials capable of ensuring reliable operation under diverse environmental conditions. The essential properties required for these materials include high thermal and robust chemical resistance, as well as favorable dielectric characteristics [1,2,3,4].

Silica cryogels produced by sol–gel chemistry represent promising candidates for advanced insulating applications due to their exceptional thermal insulation capabilities, low density, and high specific surface area attributed to their hierarchical porous structure [5]. Cryogels are typically synthesized using sol–gel methods from precursors such as sodium silicate (water glass), tetraethyl orthosilicate (TEOS), and tetramethyl orthosilicate (TMOS). Of these, water glass-based methods are widely favored due to their affordability and accessibility [5,6]. The resulting gel structure is built up by a continuous 3D solid network and a solvent phase, which is replaced by air through drying techniques, either supercritical drying (yielding aerogels) or freeze drying (cryogels). Freeze drying has gained considerable interest due to its cost-effectiveness compared to supercritical drying and because it results in high porosity. However, the non-monolithic nature of freeze-dried gels, caused by solvent crystals, may limit their practical applications [7,8,9].

Recent efforts to overcome the mechanical limitations of silica cryogels have involved developing composite materials that retain exceptional insulation properties while providing a workable monolith character. Di Luigi et al. [10] created silica cryogels reinforced with ceramic fibers, achieving thermal conductivities as low as 28 mW·m^−1^·K^−1^ and compressive strengths up to 620 kPa. Similarly, Padmanabhan et al. [11] reported silica cryogel–glass fiber blankets with low thermal conductivity (≤30 mW·m^−1^·K^−1^) and minimal bulk density (~0.05 g·cm^−3^).

Increasing environmental concerns and the demand for sustainability have led to heightened interest in biodegradable, renewable materials such as cellulose. Cellulose fibers are attractive due to their abundant availability, low cost, biodegradability, and excellent physical properties, notably, high tensile strength and outstanding natural insulating capabilities [12,13]. Wolff et al. demonstrated the potential of cellulose fibers by reinforcing bio-polyamide, achieving mechanical strengths comparable to those of conventional glass fibers at similar loadings.

In previous research, our team successfully developed biodegradable composite layers from silica cryogels and a cellulose fiber matrix [5,14]. Both starting materials are biocompatible, and the developed synthesis of composite is environmentally friendly due to its starting materials and low energy consumption. Another novel aspect of the composite is the use of a cellulose matrix rather than the typical cellulose fiber reinforcing phase.

The aim of the present research was to improve the applicability and the adsorption of the composite on distinct (smooth and rough) surfaces of various materials (from metallic to ceramic) by means of sol–gel chemistry. Our prior research characterized the adhesion of various composite formulations to different substrates. However, these systems exhibit excessive water solubility. To address this, heat treatment durations were systematically varied between 2–3 h and 72 h, with the aim of reducing water solubility. The investigation also put some effort into the enhancement of the mechanical strength of the monolith composite and the elasticity of the thin layer.

## 2. Results and Discussion

### 2.1. Characterization of the Composite’s Water Solubility

During the synthesis of the cryogel–fiber composite, partial dissolution of cellulose fibers is facilitated by the application of a NaOH–urea solution, enhancing the interactive surface area between composite constituents [15]. The cellulose fibers undergo a 30 min alkaline pre-treatment, initiating some surface dissolution. Subsequently, the fibers are mixed with hydroxyethyl cellulose (HEC) and silica cryogel particles, promoting extensive fiber–fiber and fiber–cryogel interactions due to progressive fiber dissolution [14]. Despite the good adhesion values, the initial tests have revealed significant water solubility (~97%), severely limiting practical applications. Our investigations focused on three primary parameters influencing the water solubility: the solubility of individual components (cryogel and cellulose fibers), the curing (heat treatment) duration, and residual sodium hydroxide and urea content in the composite.

Individual solubility tests confirmed that neither the cryogel nor the cellulose fibers are responsible for the dissolution. Gradually extending the curing time up to 72 h only marginally improved the solubility, reducing dissolution from 97% to approximately 89%. Further analysis identified the residual alkali content derived from the NaOH–urea treatment as the primary cause of the water solubility. Confirming this hypothesis, an additional step involving an extended fiber pre-treatment (increased from 30 min to 4.5 h) followed by thorough washing and filtration significantly reduced the residual alkali content. Layers prepared with this modified process exhibited a dramatic improvement in water solubility, dissolving only 1–3% by weight. Unfortunately, this enhanced solubility performance was accompanied by a marked reduction in adhesion properties to various substrates, including glass and metals. Thus, the next research step had to concentrate on the improvement of the layer adhesion to various surfaces.

### 2.2. Characterization of Adhesion

The reduced adhesion achieved by washing and filtration steps can be explained by the lack of NaOH. Sodium hydroxide reacts with and etches most metal surfaces, increasing the surface roughness and, thus, the mechanical interlocking between the applied layer and the metal surface. Adhesion measured on a steel surface shows a significant decrease of 37% for an alkali-free composite compared to its alkali-containing counterpart (Table 1).

A number of possible adhesion improving treatments were proposed, including treatment of the substrate with Na_2_CO_3_ and Ca(OH)_2_; increasing surface roughness through grinding with Al_2_O_3_ suspension; treatment with water glass to form a thin layer; using rough sandpaper to reduce surface smoothness; and using a combination of the alkali-containing composite in a thin primer layer, to provide the adhesion to the surface, and a second covering layer of the alkali-free sample, providing the protection against water.

The effectiveness of each proposed type of treatment was tested on a wide range of substrate surfaces, focusing on the key industrial materials including copper, steel, aluminum, and glass surfaces.

The adhesion of the alkali-free composite to steel surfaces demonstrated noticeable improvement through various applied surface treatments. The double-layer approach proved particularly effective, as the alkali-containing composite in the initial (primer) layer etches the steel surface, enhancing the physical connection and the adhesion. Water solubility tests also confirmed that the alkali-free composite applied as a protective layer effectively preserved the primer layer, maintaining low water solubility (approximately 1–3%) as long as the top layer remained intact.

The mechanical treatments such as surface roughening also significantly increased adhesion by etching the substrate surface and creating micro-roughness. The micro-roughness promotes the physical connection between the composite layer and the substrate surface.

Alkali treatment of the steel surface with Na_2_CO_3_ was carried out for durations ranging from 5 min to 1 h. The measured adhesion increased linearly up to 30 min of treatment, after which the incremental gains diminished. Thus, 30 min proved to be the optimal treatment time (Table 2).

Adhesion between the aluminum substrate surfaces and the alkali-containing sample showed a significant deterioration compared to the interaction between steel substrates and the same composite sample. This reduced adhesion is facilitated by the nature of chemical interaction between the Al_2_O_3_ layer formed on the aluminum surface and the NaOH content of the composite. The solution of Al_2_O_3_ layer results in a smooth surface. The pure, smooth aluminum surface does not support any chemical or physical interaction between the composite and surface, resulting in minimal adherence. This phenomenon is negated by applying a brief treatment using a water-glass solution, observable in the measured adhesion in Table 3, that forms a protective layer on the substrate surface, keeping the coarse Al_2_O_3_ layer, and provides an excellent point of adhesion to the composite. Removing the alkali from the sample during preparation results in an exponential growth in adhesion (Table 3). Treatment of the substrate with a weak basic solution (Na_2_CO_3_) results in the significant degradation of adhesion, which further proves all previous observations about NaOH and aluminum-oxide interaction.

Glass surfaces are glossy due to the ≡Si–O–Si≡ bonds throughout the surface. An alkali medium can modify this smooth surface through two main mechanisms. On the surface of glass, a gel-like thin layer can be easily formed, especially in humid air or a medium. The gel-like layer can be dissolved and removed by NaOH. Additionally, a strong base, like concentrated NaOH, can attack the ≡Si–O–Si≡ bonds on the surface forming silanol (≡Si–OH) groups. The Si-OH groups are capable of supporting various hydrogen bond interactions. The presence of NaOH introduces micro- and nanosized impurities on the smooth surface of the glass, resulting in a higher surface and a stronger physical connection between the sample and the substrate [16]. These three phenomena provide an explanation of the adhesion data observed in Table 4.

When using the alkali-free composite, the protective gel-like layer and the smooth surface hinder the connection between the composite and the substrate. Water glass can form a gel-like layer on the glass surface and weakens the adhesion (Table 4). The etching/physical roughening by HF, as a well-known solvent of glass, was also tested, resulting in a more course surface. The adhesion, after a single dipping in HF, shows a comparable value to that of the alkali-containing composite layer. Combining the two types of composite layers (original NaOH-containing primer layer and a layer without sodium-hydroxide content) results in the same measured adhesion as in the case of simply using the alkali-containing composite layer. Water solubility measurements reveal an additional benefit: The outer coating shields the inner primer layer, yielding an overall water solubility of just 1–3 wt%.

The observed adhesion on the copper surface and its proposed mechanism are similar to those for aluminum surfaces. Copper readily forms a copper oxide or basic copper carbonate layer on its surface with a nano-roughness. This phenomenon is observed in the results of measurements (Table 5). The adhesion value of a standard copper substrate is relatively high for an alkali-free sample (~6300 mN). The standard copper etching process uses NaOH. This also gives an explanation for the increasing measured adhesion (as seen in Table 5) when the substrate is treated with Na_2_CO_3_, or when an alkali-containing composite is applied in a thin primer layer before the application of a composite layer without sodium-hydroxide content. In both cases, the included basic etches the surface of the copper, increasing the physical interaction between the composite and the substrate surface.

Summarizing the adhesion measurements, they proved that a composite material can be applied on various surfaces effectively, including most metal surfaces (i.e., steel, copper, aluminum) and glass surfaces. The most applicable composite was prepared from silica cryogels and cellulose fibers treated for 4 h and 30 min in a sodium hydroxide and urea-containing solution. The fibers are worth filtering and washing to remove the residual alkali content.

### 2.3. Characterization of the Composite Structure

Characterization of the resulting composite system was conducted using a scanning electron microscope (SEM). A thin composite layer prepared on the surface of a steel substrate was investigated on its cross section and top layer. Some SEM results are presented on Figure 1. The left column shows the outer surface of the layer in three magnifications (100×, 1000×, and 10,000×) and the right column shows the cross section of the applied layer in the same magnifications (100×, 1000×, and 10,000×).

Lower-magnification (100×, 1000×) SEM images (Figure 1, left column) of the surface show a high level of homogeneity and macroscopic porosity. However, the higher magnification shows a microscopically smooth surface, with cellulose fibers on the surface. The average fiber size on the surface is 1.5 μm × 0.3 μm.

The composite layer cross section (Figure 1, right column) shows two distinct structures in the layer. The average thickness of the layer is 400–500 μm. The part closer to the substrate surface shows a high level of porosity with particle sizes between 1 and 6 μm. However, the outer part of the composite layer shows a high level of crystallization, with an average crystal size of 3 μm × 0.4 μm. EDX (energy-dispersive X-ray spectroscopy) was applied for determining the crystal composition, proving the growth of cellulose crystals due to heat treatment of the composite and the presence of oxygen (Figure 2).

Fourier-transform infrared spectroscopy (FT-IR) measurements confirm the formation of new bonds between the silica cryogels and the cellulose fiber matrix (Figure 3). Bands of 1366 and 1065 cm^−1^ can be found in the spectrum of pure silica and its composite. The bands are derived from silica cryogels. The 1366 cm^−1^ band represents the vibration of Si–O in the presence of Na ion. This peak is negligible in the silica gel prepared from TEOS. The band at 1065 cm^−1^ can be attributed to the stretching vibration of Si-O-Si/Na bond. The double peak at 1666 and 1619 cm^−1^ in the spectrum of cellulose and composite belongs to the cellulose. These correspond to the vibration of OH groups in water molecules absorbed in cellulose [17].

The new bands, which are not in the spectra of the starting components, are represented by “x”. The vibration band at 3453 cm^−1^ is derived from OH groups linked by H-bonds. Thus, the OH groups of cellulose and OH groups of cryogel are connected to each other by H-bonds. The other new band at 2920 cm^−1^ represents the stretch vibration of OH groups taking part in the H- bond and connected to the C atom (νC-OH). This also confirms the H-bonds between cellulose and silica.

### 2.4. Characterization of Thermal Conductivity

Table 6 presents the thermal conductivity of each component and their composites. Cellulose has one of the best thermal insulation capabilities of polymers. The thermal conductivity measurements of the alkali-free composite sample prove to be between the values measured at the individual components. There are some thermal conductivity values of electric and thermal protective polymer thin layers for comparison: polymethyl methacrylate (PMMA), 0.19–0.20 W/(m·K) [18,19]; and cellulose acetate, 0.1–0.33 W/(m·K) [20].

## 3. Conclusions

In this study, a highly protective composite thin layer has been successfully developed from sol–gel-derived porous silica cryogel particles as a dispersed phase and a pretreated cellulose matrix. In order to realize the broad potential applications of this material, particular attention was paid to optimizing the adhesion of the composite to distinct (smooth and rough) surfaces of various materials (from metallic to ceramic). Adhesion properties were measured using a specialized measurement technique developed in our laboratory. SEM analyses revealed the thickness of layers and the interactions between the matrix and the reinforcement phase as well as their morphology.

A key challenge in previous studies was the high water solubility of the composites, which can limit the commercial applicability of these materials. This solubility issue is derived from the use of a NaOH–urea solution system, which is necessary to partically dissolve the cellulose fibers, enhancing interfacial bonding with silica cryogel. The study shows that the removal of the alkali content during the preparation results in significantly lower water solubility of the composite (97 wt% → 1–3 wt%).

The reduction in NaOH content in the composite leads to a strong reduction in adhesion (e.g., it decreases on steel from 7972 mN to 5005 mN) because the alkali content can etch the smooth surfaces, improving the adhesion. Applying a treatment to the substrate’s surface—such as alkaline or acidic etching (i.e., Na_2_CO_3_, HF, water glass), mechanical roughening, or applying a thin alkali-containing primer layer—successfully increases the adhesion.

The thermal conductivity, 0.06–0.07 W/(m·K) demonstrates that the composite exhibits much better insulation properties than the protective polymer thin layers (0.1–0.4 W/(m·K)). A very important advantage of the application of these composites can be found in the environmentally friendly aspect; the thermoset polymers can be replaced by a green material. Cellulose can be characterized by good biodegradability; silica cryogel, as well as cellulose, is a biocompatible material. Thus, this composite can be considered a sufficient alternative to various polymer protective layers.

## 4. Materials and Methods

### 4.1. Synthesis of Silica Cryogels

The preparation of silica cryogels was performed by a sol–gel process followed by freeze drying. Silica gel was synthesized from sodium silicate (water glass, Na_2_SiO_3_, VWR International S.A.S., Fontenay-sous-Bois, France) precursor in an aqueous medium. The gel formation is induced by nitric acid (4 M HNO_3_, prepared from 65% HNO_3_, VWR International S.A.S., Fontenay-sous-Bois, France). Acidic conditions predominantly accelerate hydrolysis, favoring the formation of more nucleation sites with reduced branching during condensation, resulting in materials with higher porosity but smaller pore sizes [5,15]. The condensation reactions provide the 3D gel network of silica system. As previously demonstrated [5], precise control of acidic conditions significantly influences gelation kinetics, porosity, and the pore size distribution of the resulting gel structure.

During the freeze drying, the solvent (water) was replaced with air by sublimation of the frozen water crystals, effectively preserving a porous structure. Finally, the cryogel was heat treated at 500 °C to mitigate its water solubility attributed to a high density of silanol (Si–OH) groups on its large surface area [5,21].

### 4.2. Preparation of Silica Cryogel–Cellulose Fiber Composites

As previously noted, cryogels typically exhibit excellent physical properties but suffer from poor mechanical strength. The cryogels are often in existence as brittle particles. To enhance the mechanical properties of cryogels and improve their industrial applicability, our research has focused on developing composite materials from silica cryogels with cellulose fibers. In the preparation of composite, dried silica cryogel particles (<250 µm) were mixed with a solution containing NaOH (VWR International bv, Leuven, Belgium), urea (VWR International bv, Leuven, Belgium), hydroxyethyl cellulose (HEC, VWR International, Radnor, PA, USA), and viscose cellulose fibers. The NaOH–urea solution facilitates partial dissolution of the cellulose fiber surfaces [22,23,24], generating sites for effective bonding with silica cryogels. HEC is incorporated into the mixture to further enhance the bonding between the components. Silica cryogels and HEC were added to pretreated cellulose fibers. Extensive experimentation determined that a mixing duration of 4 h optimally balances component interaction and processing efficiency. After mixing, the solvent of the composite solution was evaporated, followed by a drying process at 120 °C.

### 4.3. Characterization

The *morphology* and *structure characteristics* of the composite materials were examined using a field emission scanning electron microscope (FE-SEM, FEI Quanta 3D FEG, FEI Company, Hillsboro, OR, USA). The SEM system was equipped with energy-dispersive X-ray spectroscopy (EDX) for elemental analysis. To obtain high-contrast imaging, the samples were deposited onto a highly ordered pyrolytic graphite (HOPG) substrate. Particle size and fiber thickness were quantitatively assessed using the ImageJ software (Version 1.54p, National Institutes of Health, Bethesda, MD, USA), while porosity analysis was conducted using GIMP (Version 2.10.36, The GIMP Development Team, https://www.gimp.org/), which facilitated image-based segmentation and thresholding techniques.

*ATR-FTIR* spectra were acquired on a Bruker IFS 55 spectrometer equipped with a diamond ATR accessory (PIKE Technologies, PIKE Technologies, Fitchburg, WI, USA). Measurements were made over the 4000–500 cm^−1^ spectral region at a resolution of 1 cm^−1^. Samples were pressed directly against the ATR crystal, yielding an effective penetration depth of approximately 0.5–2 μm. A Globar source and a DTGS detector were used for all experiments.

*Thermal conductivity* measurements were performed using a heat-flow meter apparatus (Holometrix, Inc., formerly Dynatech R/D Company, Cambridge, MA, USA) to evaluate the insulating properties of the fibers, the cryogel, and their composite material. Calibration of the instrument was verified against a reference sample of CWS glass wool with a known thermal conductivity of 0.032 W/(m·K). The effective thermal conductivity was recorded over a temperature range of 20–100 °C at five distinct measurement points. The sample area within the testing zone was 10 × 10 cm, ensuring consistency in thermal diffusion analysis.

The *adhesion strength* between the composite layer and various substrates was evaluated using a testing apparatus developed in our laboratory. The setup incorporated a YZC-133-SCL-5 strain gauge sensor, which was integrated with a load cell (600 × 127 × 127 mm) to quantify the force necessary to detach the layer from the substrate. Multiple independent tests confirmed that the required force was unaffected by the thickness of the applied layer. During testing, a force was applied along the +*z* axis, inducing tensile stress within the strain gauge and generating an internal shear force. The resulting analog signal was processed using an HX711 analog-to-digital (AD) converter, which relayed the data to an Arduino Uno programmable logic controller (PLC) for real-time force measurement. To ensure accuracy, the system was calibrated using three reference weights (187 g, 336 g, and 525 g), each tested in triplicate. The computed calibration factor for the measurement system was determined to be 2.127. The baseline test value was defined as 0 adhesion, measured on the Teflon surface.

## Figures and Tables

**Figure 1 gels-11-00522-f001:**
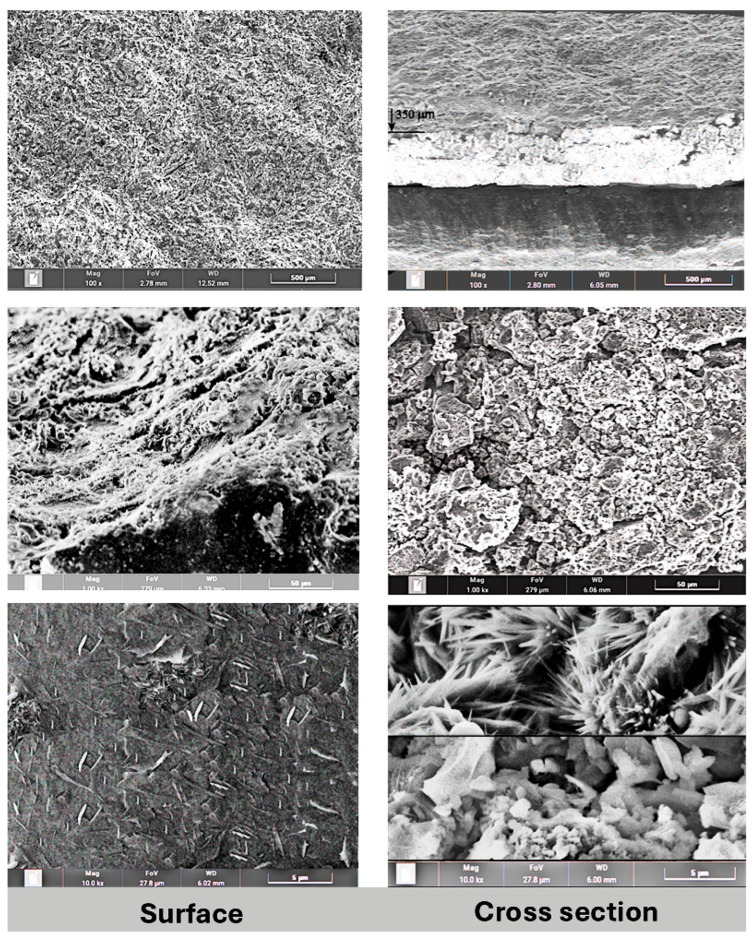
SEM micrographs of a composite sample prepared on a steel substrate, showing the top surface of the layer (**left** images) and cross section of the composite layer (**right** images).

**Figure 2 gels-11-00522-f002:**
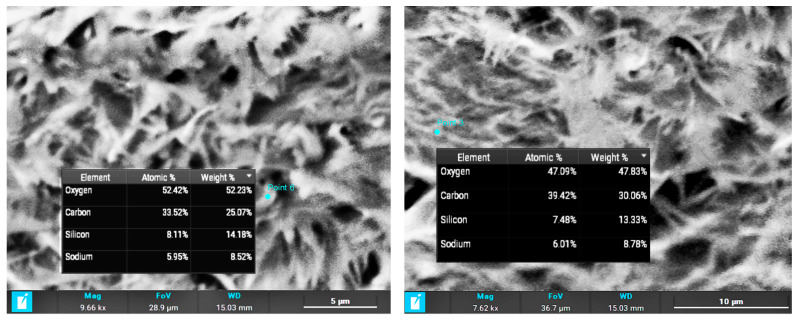
Crystal composition measured using EDX.

**Figure 3 gels-11-00522-f003:**
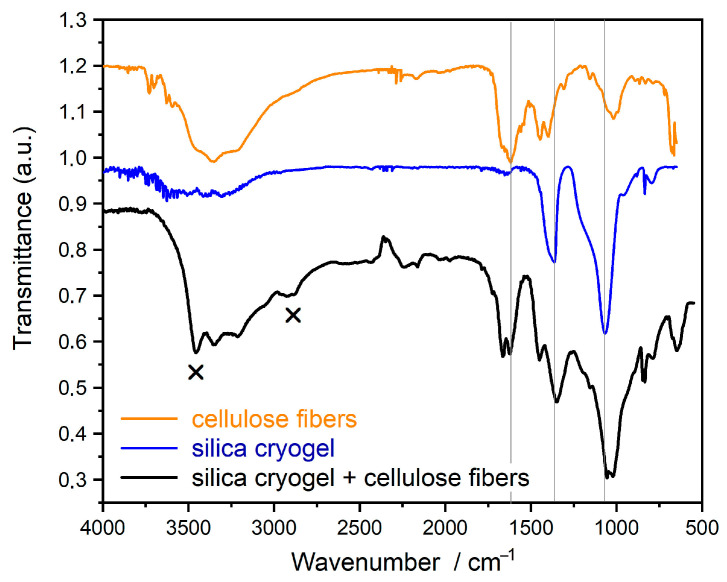
FTIR spectra of single components (cellulose and silica cryogel) and their composite.

**Table 1 gels-11-00522-t001:** Adhesion difference between the original composite sample and the composite sample with reduced NaOH–urea content.

Composite Material	Measured Adhesion/mN
Alkali-containing composite	7972 ± 200
Alkali-free composite	5005 ± 100

**Table 2 gels-11-00522-t002:** Adhesion measurement results tested on a steel surface. Comparison of the alkali-containing composite sample, new samples prepared by removing the alkali from the system, and the samples treated by different surface treatment methods.

Material	Treatment Type	Treatment Time/min	Measured Adhesion/mN
Alkali-containing sample	-	-	7972 ± 200
Alkali-free composites	-	-	5005 ± 200
	Grinding by Al_2_O_3_ suspension	-	5509 ± 100
	Water glass	10	7428 ± 200
	Roughening with sandpaper	-	
	Na_2_CO_3_	30	8137 ± 200
	Primer alkali-containing composite thin layer	-	8701 ± 200

**Table 3 gels-11-00522-t003:** Adhesion measurement results comparison of the original composite sample containing alkali, the new sample prepared by removing the alkali from the system, and different surface treatment methods, tested on an aluminum surface.

Material	Treatment Type	Treatment Time/min	Measured Adhesion/mN
Original alkali-containing sample	-	-	0
	Water glass	10	5174 ± 100
Alkali-free composite	-	-	7448 ± 200
	Na_2_CO_3_	30	5070 ± 100

**Table 4 gels-11-00522-t004:** Adhesion measurement results comparison of the original composite sample containing alkali, a new alkali-free composite prepared by removing the alkali from the system, and different surface treatment methods, tested on a glass surface.

Material	Treatment Type	Treatment Time/min	Measured Adhesion/mN
Original alkali-containing sample	-	-	11,413 ± 250
Alkali-free composite	-	-	4256 ± 50
	Water glass	10	5143 ± 50
	HF	0.2	10,271 ± 250
	Thin primer layer of alkali-containing sample	-	10,931 ± 250

**Table 5 gels-11-00522-t005:** Adhesion measurement results comparison of the original composite sample containing alkali, the new sample prepared by removing the alkali from the system, and different surface treatment methods, tested on the surface of a copper substrate.

Material	Treatment Type	Treatment Time/min	Measured Adhesion/mN
Original **alkali-containing sample**	-	-	14,540 ± 250
Composite prepared by the removal of NaOH during the prep (**alkali-free composite**)	-	-	6280 ± 100
	Na_2_CO_3_	30	7407 ± 200
	Thin layer of alkali-containing composite	-	16,901 ± 300

**Table 6 gels-11-00522-t006:** Thermal conductivity measurements of the alkali-containing composite and the individual components.

Temp. (°C)	Thermal Conductivity, λ (W/(m·K)			
	Cellulose Fibers	SiO_2_ Cryogel	Alkali-Containing Composite	Alkali-Free Composite
20	0.087	0.033	0.068	0.064
50	0.089	0.036	0.070	0.062
100	–	0.041	0.071	0.063

## Data Availability

The original contributions presented in this study are included in the article. Further inquiries can be directed to the corresponding author.

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
