# Peer review of "Biocompatible Composite Protective Thin Layer Containing Cellulose Fibers and Silica Cryogel"

_gels, 2025, doi:10.3390/gels11070522_

Round 1

Reviewer 1 Report

Comments and Suggestions for Authors

Dear, Authors

I am pleased to review you manuscript. The manuscript is very interesting. 

I would like to give some suggestions.

  1. The title of this manuscript present the Biocompatible but in the context did not show the result. If possible please add some result that can be relate to the biocopatible.
  2. In the Results and Discussion part have context about the SEM with EDX experiment. If possible please show the result about composition of materials by EDX. 

Author Response

We would like to thank Reviewer #1 for their insightful and constructive comments, which have helped us improve the quality of our manuscript. We have carefully revised the paper in response to your suggestions. Please see the attachment for our detailed response to each comment/question.

Reviewer 2 Report

Comments and Suggestions for Authors

This is a paper about synthesizing protective composite layers from biodegradable cellulose and biocompatible, sol-gel-derived silica cryogel. However, several points are important to be addressed before going to a possible publication in this journal.

  • The title of the manuscript is general, It is better to mention the main components in the title.
  • The abstract does not follow the scientific format, which includes background, methods, quantitative results, and conclusion.
  • The exact weight percentages of cellulose, silica, NaOH, HEC, etc., are not provided.
  • Characterization techniques such as XRD, FT-IR, or TGA for thermal stability should be provided.
  • Experimental procedures and formulations were unclear.
  • Some references are presented with complete citation details while others are abbreviated or follow a different formatting style.
  • The mechanism of NaOH interaction with metal surfaces or the reduction of silanol groups through heat treatment is presented without proper citation.

Comments on the Quality of English Language

The overall manuscript must be checked in terms of grammar, punctuation, and syntax. There are many spelling and grammatical errors in the article.

Author Response

We would like to thank Reviewer #2 for their insightful and constructive comments, which have helped us improve the quality of our manuscript. We have carefully revised the paper in response to your suggestions. Please see the attachment for our detailed response to each comment/question.

Reviewer 3 Report

Comments and Suggestions for Authors

Review

Biocompatible Composite Protective Thin Layer Using Sol-gel Chemistry

This article details the development of a novel composite material – a biodegradable cellulose matrix reinforced with silica cryogel – designed for protective coating applications. The research successfully addresses the critical issue of water solubility. A key strength of this material lies in its thermal and electrical insulation properties, alongside its biocompatibility and potential for environmental sustainability. The study’s novelty resides in the combination of these eco-friendly materials and the tailored surface treatments to enhance adhesion to diverse substrates.

The research is well-designed, the methods used are adequate, the results are good, however the paper’s writing has to be improved.

I recommend the correction of the paper according to the following observations and questions.

Abstract

- The abstract does not clearly state what the composite is protecting against. Is it thermal stress, corrosion, electrical currents, or something else?

- An abstract should not only state what was done, but also summarize the key results and whether the research goals were met. Did the researchers successfully create a material with the desired protective properties?

Introduction

- The text mentions applying the composite to "distinct (smooth and rough) surfaces of various materials (from metallic to ceramic)." Why is adhesion to these surfaces a significant challenge? What existing solutions are inadequate? The introduction should frame this as a significant problem the research aims to solve, not just a characteristic of the research.

- Water solubility is a crucial limitation that impacts practical applicability. The description of "extensive experiments...in order to reduce the water solubility" seems like damage control rather than a proactive solution. The stated aim ("to improve the applicability…by means of sol-gel chemistry") is too broad. What specific applications are targeted? “Applicability” is not a measurable outcome.

- Why is cellulose incorporated specifically with the cryogel, beyond general sustainability concerns. Is it for cost? Mechanical properties? A specific application need? What advantages does this matrix approach offer over typical fibre reinforcement? This needs to be supported by some explanation.

- The wide range of heat treatment durations (2-3 hours to 72 hours) is presented in the introduction without explanation. What is the presumed effect of heat treatment on water solubility, mechanical strength, elasticity?

- “Cryogels are typically synthesized using sol-gel methods from precursors such as sodium silicate (water glass), tetraethyl orthosilicate (TEOS), or tetramethyl orthosilicate (TMOS). Of these, water glass-based methods are widely favored due to their affordability and accessibility [5, 6]” - this statement might be true for silica cryogels, but not in general for any cryogel. Expand the discussion to acknowledge that cryogels can be synthesized from a wider range of materials, including biopolymers like gelatin, alginate, and polyacrylamide. Include citations relevant to these non-silica cryogels.

- Enhancement of mechanical strength and elasticity: What level of enhancement is being targeted? What are the desired mechanical properties (e.g., tensile strength, Young's modulus)?

- "Prior research established various adhesions": This needs to be more specific. What surfaces were tested? What adhesion strengths were achieved? Were these measurements reproducible?

Materials and methods

- This part is lacking detailed information needed for reproducibility. There is no information, for example, about reagent concentrations, precise process parameters, cellulose fibre details, gelation time, freezing rate etc. Adding this information will significantly strengthen the scientific rigor of the study.

- How were the samples prepared for thermal conductivity measurements (e.g., thickness, density, surface size)?

- It would be clearer to list the substrate materials and the method of application of the layer in this subchapter, that were used for adhesion testing

- What was the rate at which the force was applied during the adhesion tests?

- How many replicates were performed for the adhesion tests? What statistical analysis was used to determine the significance of the results? The section lacks any mention of error analysis or uncertainty estimates.

- Why was mechanical testing not applied?

Discussion

- SEM/EDX was used to confirm crystal composition and the presence of oxygen. What elements were detected, and what does this indicates about the crystallization process? Were there any changes in Si:O ratios, for example?

- Were there any control samples (e.g., a substrate with no treatment) to give a baseline adhesion value?

- Were any long-term stability tests conducted to assess the durability of the adhesion over time and under various environmental conditions?

- How was water solubility measured (weight loss, visual observation)?.

Conclusions

- While the reduction in adhesion due to alkali removal is mentioned, a more detailed summary of how the surface treatments successfully overcame this reduction would be beneficial.

- A brief mention of potential future research directions (e.g., long-term durability testing, scale-up of production, exploring different cellulose sources) could add further value.

- The conclusion mentions a "specialized measurement technique" for adhesion. While this highlights the novelty of the approach, it would be stronger if briefly mentioned what makes it specialized or why a standard technique wasn’t suitable.

Comments on the Quality of English Language

Stylistic errors

The text contains many stylistic and also grammatical errors, as well as imprecise composition of sentences.

For example:

“Our prior research established various adhesions to different surfaces.“ (61)

This sentence contains several grammatical and stylistic errors:

  • “Adhesion” is usually uncountable when referring to the phenomenon or property of sticking
  • The plural "adhesions" is typically used in medical contexts, not in materials science or surface interactions
  • "Various adhesions to different surfaces" is redundant and imprecise.
  • Consider being more specific: adhesion to a range of surface types, or substrates.

Suggested: Our earlier studies investigated adhesion properties on different surface materials.

I suggest takeing into account a correction tool or the help of a native English-speaking colleague.

Author Response

We would like to thank Reviewer #3 for their insightful and constructive comments, which have helped us improve the quality of our manuscript. We have carefully revised the paper in response to your suggestions. Please see the attachment for our detailed response to each comment/question.

Round 2

Reviewer 1 Report

Comments and Suggestions for Authors

About the title of the manuscript can be misleading it will be great if to modify for clear understand.

Author Response

We thank the reviewers for their valuable comments that have permitted us to improve the quality of our paper. We have revised the manuscript accordingly.

Biocompatible composite protective thin layer based on cellulose and silica cryogel

Reviewer 2 Report

Comments and Suggestions for Authors

-

Author Response

Thank you for your review. As there were no additional comments or suggestions regarding the content of the manuscript, no further changes were made. To ensure the quality of the language, the manuscript was reviewed by a native English speaker for grammar and sentence structure.